# Review of Non-Eosinophilic Esophagitis-Eosinophilic Gastrointestinal Disease (Non-EoE-EGID) and a Case Series of Twenty-Eight Affected Patients

**DOI:** 10.3390/biom13091417

**Published:** 2023-09-20

**Authors:** Yoshikazu Kinoshita, Tsuyoshi Sanuki

**Affiliations:** Department of Medicine and Gastroenterology, Hyogo Prefectural Harima-Himeji General Medical Center, Himeji 670-8560, Japan

**Keywords:** eosinophil, allergy, gastrointestinal tract, esophagus, functional dyspepsia, irritable bowel syndrome

## Abstract

Eosinophilic gastrointestinal disease (EGID) is divided into eosinophilic esophagitis (EoE) and non-eosinophilic esophagitis eosinophilic gastrointestinal disease (non-EoE-EGID) based on the involved gastrointestinal segments. Reports regarding non-EoE-EGID are limited, in part because of its rarity. The present study was performed to review non-EoE-EGID, including its pathogenesis, diagnosis, treatment, and prognosis. Additionally, details regarding 28 cases of non-EoE-EGID recently diagnosed at our Japanese tertial medical center are presented and compared with 20 EoE cases diagnosed during the same period at the same medical center. Comparisons of the two groups clarified differences regarding age- and gender-dependent prevalence between the two conditions, and also showed that systemic involvement and disease severity were greater in the non-EoE-EGID patients. Notably, diagnosis of non-EoE-EGID is difficult because of its lack of specific or characteristic symptoms and endoscopic findings. The clinical characteristics of EoE and non-EoE-EGID differ in many ways, while they also share several genetic, clinical, laboratory, and histopathological features.

## 1. Introduction

Eosinophilic gastrointestinal disease (EGID) is defined as a condition with gastrointestinal symptoms and pathologically dense infiltration of eosinophils in the gastrointestinal tract. Affected patients are divided into eosinophilic esophagitis (EoE) cases, with eosinophil infiltration only in the esophagus, and non-eosinophilic esophagitis eosinophilic gastrointestinal disease (non-EoE-EGID) cases, with gastrointestinal eosinophil infiltration irrespective of esophageal involvement [1]. Non-EoE-EGID is also used to describe eosinophilic gastritis, eosinophilic enteritis, eosinophilic gastroenteritis, and eosinophilic colitis cases, depending on the involved segments of the gastrointestinal tract. EoE and non-EoE-EGID share a similar pathogenesis, as both conditions have been found in the same family members and even in the same individuals [2]. In addition, transcriptome analysis has shown similarities between EoE and non-EoE-EGID, although important differences between eosinophilic gastritis and eosinophilic colitis have also been found [3,4].

The prevalence of EoE has been reported to be approximately 50 in 100,000 of the general population, while that of non-EoE-EGID is considered to be less than 10 in 100,000 of the general population [5,6]. On the one hand, because of the higher prevalence rate and also the homogeneity of reported cases, basic and clinical research studies of EGID have mainly focused on EoE. On the other hand, basic information and clinical findings regarding non-EoE-EGID are limited in the literature because of its rarity and the heterogeneity of affected patients. Notably, diverse findings related to non-EoE-EGID have been observed in different segments and layers of the involved gastrointestinal tract [3,7].

In this paper, a review of factors related to the pathogenesis, epidemiology, diagnosis, and treatment of non-EoE-EGID is provided in a narrative and concise fashion. In addition, a summary of 28 cases of non-EoE-EGID recently diagnosed at a single tertial medical center is presented. Furthermore, the clinical characteristics of the 28 non-EoE-EGID cases are compared with those of 20 EoE cases diagnosed at the same medical center during the same period. 

## 2. Review

### 2.1. Pathogenesis of EoE and Non-EoE-EGID

Investigations of the pathogenesis of EoE have been performed and reports of its core mechanism have been presented. For tissue entry by allergens, damage to the esophageal squamous epithelium causing increased permeability is necessary and also release of alarmins, including IL-25 and -33, and thymic stromal lymphopoietin (TSLP), resulting in type 2 innate lymphoid cell (ILC2) and T helper type 2 (Th2) cell activation [8,9]. Gastric acid refluxed from the stomach causing damage to the esophageal squamous epithelium is considered to be an important factor. As a result, EoE is most frequently found in young males with higher levels of gastric acid secretion, and associated lesions are most frequently formed on the distal area of the esophagus [10]. Vonoprazan, a potassium competitive acid blocker, as well as proton pump inhibitors are effective for treatment of EoE, with greater than 50% of affected patients reported to be successfully treated by these acid inhibitors, possibly because of their protective effects against acid-induced damage and increased permeability of esophageal mucosa [11]. Food and airborne allergens, including wheat, milk, egg, and pollen, penetrate the esophageal epithelium, enter mucosal tissue, and activate mast cells and eosinophils through stimulation of ILC2 and Th2-type lymphocytes. In EoE cases, IL-5, -13, and eotaxins are key molecules involved in immune reactions, which result in increased TGF beta and periostin, leading to fibrosis of the esophagus (Figure 1). Recently, esophageal sensitization to pollens has been reported to be important as a trigger and a sustaining factor of esophageal eosinophilia [12].

Factors related to the pathogenesis of non-EoE-EGID have yet to be fully elucidated. It has been reported that family members of EoE patients have an increased risk of developing non-EoE-EGID as well as EoE, and it is speculated that the two diseases have similar genetic backgrounds. Furthermore, 10–30% of non-EoE-EGID patients also develop pathological esophageal eosinophil infiltration [2]. Indeed, transcriptome analysis findings have indicated some similarities of EoE with eosinophilic gastritis, while the similarity between EoE and eosinophilic colitis is limited [3,4]; the research results suggest a similar pathogenesis for non-EoE-EGID and EoE, and also indicate, in part, that the heterogeneity of the pathogenesis of non-EoE-EGID is dependent on the involved gastrointestinal segments. Eotaxin 3 is elevated in both EoE and eosinophilic gastritis, while eotaxin 1 is elevated in eosinophilic colitis. Allergen modifications caused by digestive enzymes and the gut microbiome as well as bioactive substances produced by the gut microbiome are also related to the complexity of non-EoE-EGID (Figure 2) [13]. Further research is necessary to understand the full pathogenesis of non-EoE-EGID cases.

### 2.2. Diagnosis of Non-EoE-EGID

#### 2.2.1. Epidemiology

Research regarding the epidemiology of non-EoE-EGID in Western countries has indicated a lower prevalence compared to that of EoE, i.e., less than 10 cases in 100,000 of the general population. However, some investigators have suggested that approximately 2% of symptomatic patients thought to be affected by gastrointestinal disease may actually be affected by non-EoE-EGID [14,15]. A large number of patients with non-EoE-EGID are not appropriately diagnosed at their first visit and diagnostic delay has been found in many cases, thus the prevalence may actually be higher [16]. Education for physicians in this regard and studies performed to determine precise rates of incidence and prevalence of non-EoE-EGID will be necessary. Additionally, non-EoE-EGID family clustering is rarely reported. Indeed, in the series of cases reported later in this study, there was no such incidence noted. 

#### 2.2.2. Symptoms

The symptoms reported by patients are dependent on the involved gastrointestinal tract segments. Patients with gastroduodenal lesions frequently note epigastralgia and nausea/vomiting, whereas patients with ileo-colonic lesions frequently report lower abdominal pain and diarrhea [17,18]. In the present case series, 78% of the non-EoE-EGID patients complained of abdominal pain and 50% of the patients reported diarrhea, while those with accompanying esophageal lesions often noted dysphagia and heartburn. Symptoms reported by patients with non-EoE-EGID may be similar to those affecting patients with functional dyspepsia and irritable bowel symptoms [16,19]. Since the symptoms associated with non-EoE-EGID are non-specific, a patient with abdominal symptoms without a known cause might be suspected as possible non-EoE-EGID, especially when they have an accompanying atopic disease.

#### 2.2.3. Laboratory Testing

Elevated eosinophil counts in peripheral blood have been found in 70–80% of non-EoE-EGID cases, higher than in cases of EoE, which was confirmed in the present case series [18]. Elevated IgE level has also been found in approximately 50% of non-EoE-EGID patients, while elevated CRP and decreased albumin concentration have been reported to be found in 20–30% of non-EoE-EGID patients. These laboratory test findings suggest that systemic involvement is more severe in non-EoE-EGID as compared to EoE. Serum anti-Helicobacter pylori antibody positivity has been found to be lower in cases with non-EoE-EGID than in a control group [20]. Unfortunately, no specific sensitive blood biomarker has been reported for diagnosing adult cases with non-EoE-EGID or severity assessment, although some non-invasive and minimally invasive biomarkers for EoE are under investigation [21,22,23,24]. Fecal calprotectin has been reported to be increased in cases with EGID and may be a screening marker for EGID [24]. Increased fecal eosinophil granular proteins, such as eosinophil cationic protein, may be diagnostic markers indicating EGID, although more results are needed to confirm their usefulness [23,25].

#### 2.2.4. Diagnostic Imaging

Computed tomography and ultrasonographical examination findings can reveal gastrointestinal tract segmental thickening and the presence of ascites [17]; however, an imaging study might also find no specific abnormality in affected patients.

#### 2.2.5. Endoscopy

An endoscopic examination can reveal various non-specific abnormalities, including mucosal redness, edema, erosion/ulcer, granularity, and nodules, although such abnormalities can be found in a variety of gastrointestinal diseases as well. In addition, on the one hand, in approximately 60% of investigated cases with non-EoE-EGID, no abnormal endoscopic finding was noted [26,27]. These results suggest that an endoscopic examination has limited value for diagnosing non-EoE-EGID. On the other hand, endoscopy is known to be sensitive enough for diagnosing EoE, since multiple specific abnormalities, such as longitudinal furrows, rings, and white plaque, can be detected [28,29]. Nevertheless, endoscopy is useful for obtaining appropriate biopsy samples for a histopathological examination in suspected EoE as well as non-EoE-EGID cases.

#### 2.2.6. Histopathological Examination

Patients with non-EoE-EGID have increased eosinophil infiltration in gastrointestinal tissue. No eosinophils can be found in esophageal stratified squamous epithelium of normal healthy individuals; thus, their presence in the esophageal epithelial layer should be considered abnormal. However, in other portions of the gastrointestinal tract covered by single-layered columnar epithelium, eosinophil infiltration can be found even in healthy individuals [30]. The density of mucosal eosinophil infiltration has large varieties in different segments of the gastrointestinal tract. In healthy individuals, eosinophil infiltration is highest in the distal ileum and right side of the colon, while it is lower in both the oral and anal sides [30,31]. Eosinophil infiltration greater than 20/high-power field (HPF) (×400) has been reported to be found in the right side of the colon in healthy individuals, different from the esophageal epithelial layer. International consensus concerning the upper cut-off limit of normal eosinophil infiltration has yet to be reached and no international guidelines have been published. Interobserver variance of eosinophil counts in gastrointestinal mucosal biopsy specimens has been reported to be not good enough [32]. Furthermore, the visual size of the microscopic HPF differs among types of microscopes, with a nearly four-times difference in some cases, which may cause difficulties with determining the eosinophil count for a diagnosis of EGID. As a result, cut-off levels for a diagnosis of non-EoE-EGID differ among investigators and indicate the need for diagnostic markers other than simple density of eosinophil infiltration [30,31,32]. Eosinophil degradation and the presence of intra-epithelial eosinophils, suggesting eosinophil activation, as well as epithelial necrosis may be useful factors for a non-EoE-EGID diagnosis [33,34]. Recent studies that used single-cell transcriptome analysis have noted the heterogeneity of eosinophils infiltrating tissues; thus, indicating the necessity of markers of eosinophil activation for appropriate diagnosis of non-EoE-EGID and severity assessment [35].

When performing a histopathological diagnosis, appropriate tissue samples are necessary. For sensitivity to provide data for specific diagnosis, it has been reported that more than five biopsy samples are necessary, since tissue eosinophil infiltration is not homogenous but rather patchy [36]. Tissue samples for diagnosing non-EoE-EGID do not necessarily need to be taken from endoscopically identified lesions, which is different from suspected EoE cases [37]. In EoE, eosinophil density is reported to be higher in endoscopically identified lesions, including longitudinal furrows and white plaque areas, while in non-EoE-EGID cases, the sensitivity of targeted and random biopsy examinations has been reported to be the same [37,38]. Thus, endoscopically identified lesions may not indicate those with dense eosinophil infiltration.

#### 2.2.7. Strategy for Diagnosing Non-EoE-EGID

Non-EoE-EGID should be suspected in patients with unexplained gastrointestinal symptoms, especially if they have atopic disease as well. While laboratory testing and endoscopic examinations have limited diagnostic value, 70–80% of non-EoE-EGID cases show peripheral blood eosinophilia. Even in the absence of an endoscopic abnormality, at least five biopsy specimens should be taken from each suspected gastrointestinal segment [7,36]. The attending pathologist must be informed that non-EoE-EGID is suspected in the investigated case, and a quantitative evaluation of eosinophil infiltration will be necessary [39].

The simultaneous presence of unexplained gastrointestinal symptoms and pathologically identified dense eosinophil infiltration is not necessarily enough to confirm a diagnosis of non-EoE-EGID, as there are several diseases that should also be considered for differential diagnosis, including ulcerative colitis, Crohn’s disease, and celiac disease (Table 1) [40]. This process is complicated and time-consuming, but important for a correct non-EoE-EGID diagnosis.

### 2.3. Treatment for Non-EoE-EGID

Several different treatment options for non-EoE-EGID have been reported, although evidence for the effectiveness of each is not adequate. Molecular targeting therapy using a specific antibody is under development for non-EoE-EGID patients, such as anti-IL-4/13R antibody treatment, currently available for EoE cases [41,42].

#### 2.3.1. Dietary Therapy

An elimination diet and elemental diet without allergens are theoretically most appropriate, when allergens can be identified in the patient. Test results for the anti-allergen IgE antibody are not adequately specific or sensitive for an elimination diet, although possible involvement of IgE in development of non-EoE-EGID has been suggested [43]. Since skin prick and patch test results are not adequately sensitive for detecting pathogenic allergens, use of an empirical elimination diet has been tested with various results presented. According to a recently reported meta-analysis including over 1700 cases, the efficacy of empirical elimination diets for EoE is higher than that of targeted elimination diets [44]. While several case reports and case series results have suggested the value of an empirical elimination diet for non-EoE-EGID as well as for EoE, the primary endpoints of the reported studies were mainly subjective symptoms and not histopathologically identified improvement of eosinophil-related inflammation. We have reported a patient with non-EoE-EGID who was treated by the empirical elimination diet [45]. In this case, histopathological as well as symptomatic improvement was confirmed, and the value of empirical elimination diet was suggested. In addition, a recently reported study also suggested the effectiveness of empirical elimination diet for non-EoE-EGID not only by the change of subjective symptoms but also by the objective tests including blood eosinophil count, serum albumin, as well as thymus and activation-regulated chemokine (TARC). The results of the study showed that six of seven tested cases had some improvement of these objective tests [46]. Elemental, 6/7 food elimination diets, and cow milk elimination therapy have been reported to achieve at least clinical improvement in 75.8%, 85.3%, and 62%, respectively, of treated patients [47,48,49]. Although theoretically useful, the elimination diet for patients with non-EoE-EGID is still a developing treatment and there is a need for clinical research.

#### 2.3.2. Proton Pump Inhibitor Therapy

Use of proton pump inhibitors (PPIs) has been reported to be effective for treatment of EoE, in part because of the anti-Th2 inflammatory effect and also through suppression of gastric acid-induced increased mucosal permeability of the distal esophagus [50,51]. Although an anti-inflammatory effect of PPIs on non-EoE-EGID has yet to be confirmed, PPI treatment is expected to suppress acid-related aggravation of gastro-duodenal lesions, such as erosion and ulcers caused by non-EoE-EGID. Different from EoE, the evidence clearly showing the effectiveness of PPI for non-EoE-EGID is not available.

#### 2.3.3. Leukotriene-Receptor Antagonist

Montelukast sodium has been used for the treatment of non-EoE-EGID patients and its beneficial effects were shown in a small-sized double-blind study [52]. Although its adverse effects are limited, potency is limited; thus, montelukast is mainly used for non-severe cases as well as for patients undergoing a regimen to decrease systemic glucocorticoid dose.

#### 2.3.4. Antihistamines, Cromoglycate, and Suplatast Tosilate

Antihistamines, cromoglycate, and suplatast tosilate are anti-allergic drugs that suppress the effects of histamine, stabilize mast cells, and inhibit Th2-type immune reaction, respectively. They are widely given for atopic diseases and may also be effective for non-EoE-EGID, although no known appropriate study has been conducted to investigate their effectiveness. However, results presented in several case reports and case series have suggested good effects for non-severe cases [53]. Different from EoE, effective treatment options for non-EoE-EGID are limited and clinical research studies are needed to investigate the roles of these drugs for treatment of non-EoE-EGID patients.

#### 2.3.5. Systemic Glucocorticoid

Prednisolone is the most widely employed medication for treatment of non-EoE-EGID cases, although no double-blind study that investigated the usefulness of this systemic glucocorticoid has been presented. When a dose of 30–40 mg/day has been administered for non-EoE-EGID, clinical and histopathological remission have been achieved, at least temporarily, in the majority of patients [17,54]. Furthermore, in another study, 42% of patients continued in a state of remission even after dose reduction or termination, although 37% of patients showed repeated relapse of the disease during the dose reduction phase. To prevent relapse, continuous administration of prednisolone was frequently necessary. In 21% of the cases, prednisolone administration failed to suppress disease activity adequately to maintain the remission state [55]. Neither an appropriate dose of prednisolone for remission induction nor a standard dose-reduction strategy has been established.

#### 2.3.6. Topical Glucocorticoids

Administration of a topical glucocorticoid, which rapidly becomes degraded during the first pass through the liver, can be used to reduce the risk of adverse effects of glucocorticoid therapy. Budesonide is frequently given as a topical glucocorticoid with beneficial effects, and it has received approval and is clinically employed for treatment of EoE [56,57]. However, its effects for non-EoE-EGID have only been noted in case reports with lower grade evidence. Additional studies conducted to investigate the effects of topical glucocorticoids including fluticasone and budesonide are needed.

#### 2.3.7. Immunomodulators

When the dose of prednisolone cannot be reduced after long-term treatment, azathioprine and 6-mercaptopurine may be useful. Several case reports have shown beneficial effects of these drugs for reducing prednisolone dose, although no clinical study with a high level of evidence has been presented [58,59].

#### 2.3.8. Biological Agents

Several specific antibodies against key molecules related to development of non-EoE-EGID are under development. Results of a phase 2 study of the effects of lirentelimab (anti-Siglec-8 antibody) indicate that its administration results in decreased tissue eosinophil infiltration and relieves symptoms in patients with non-EoE-EGID [60]. Additionally, the effects of anti-IL-13 antibody treatment on non-EoE-EGID are now under investigation.

Several different types of treatment have been employed for non-EoE-EGID, although many are used in a traditional manner without adequate evidence of effectiveness. In clinical practice, on the one hand, anti-allergic drugs including leukotriene-receptor antagonist tend to be selected for non-EoE-EGID with milder inflammation as the first line drugs. Prednisolone, on the other hand, tends to be administered to non-EoE-EGID with more severe inflammation as the first line drug. When these medications are not effective, consensus is not reached on the additional treatment strategy. Safer and more effectual treatments are under development. The individual roles of the various treatments available for remission induction and maintenance therapy in non-EoE-EGID cases have yet to be fixed, as their benefits and the adverse effects are not fully clarified. Additional studies are necessary to construct an effective treatment strategy and treatment flow, in addition to the development of modern therapies.

### 2.4. Prognosis of Non-EoE-EGID

Different from EoE, the prognosis of non-EoE-EGID patients is not adequate at this time. Approximately, half of affected patients require continuous or intermittent long-term administration of a systemic glucocorticoid. However, such long-term administration is associated with various adverse effects, thus requiring additional treatments for drug-related complications. Therefore, compared with EoE patients, those with non-EoE-EGID require additional medical resources and the overall cost for therapy has been reported to be higher [61,62].

Another important factor is the life expectancy of patients with non-EoE-EGID, especially eosinophilic gastritis, which is known to be shorter than that of healthy individuals [63,64]. In addition to drug-related complications, neoplasia possibly caused by long-term inflammation and cardiovascular complications associated with high-grade eosinophilia may be reasons for the reduced life expectancy of affected patients. Development of more effective and safer treatment strategies for treating non-EoE-EGID is necessary.

## 3. Case Series

### 3.1. EoE and Non-EoE-EGID Cases

All the consecutive patients with EGID treated at our department from April 2019 to March 2023 were included in this case series. In this retrospective study, 28 patients with non-EoE-EGID and 20 patients with EoE were included (Table 2). A thorough medical history was taken, and a physical examination was performed, for each patient. Blood and urine laboratory tests were also done in all the cases. In addition, all the cases were investigated by upper gastrointestinal endoscopy with esophageal, gastric, and duodenal mucosal biopsies. When a patient complained of lower gastrointestinal symptoms including diarrhea or when their physical examinations suggested lower abdominal abnormalities, a colonoscopy with multiple biopsies was also performed. Patients with EoE were confirmed not to be complicated with non-EoE-EGID. Since this is a retrospective case series, however, not all the gastrointestinal segments of included patients were investigated histologically.

The mean age of those with non-EoE-EGID was 44 years, with a wide distribution from teenagers to elderly aged 80 years and older. The EoE patients were mainly in their 40s and 50s, with a mean age of 47 years. For the non-EoE-EGID cases, 16 (57%) of the 28 patients were female (Figure 3), a preponderance often reported by others. In addition, male preponderance was noted in the cases with EoE (13 of 20, 65%). Fourteen patients (50%) with non-EoE-EGID had atopic comorbidities, including asthma (*n* = 6), allergic rhinitis (*n* = 5), food allergy (*n* = 4), eosinophilic chronic rhinosinusitis (*n* = 3), atopic dermatitis (*n* = 2), drug allergy (*n* = 2), and eosinophilic otitis media (*n* = 1 case), including some with multiple atopic comorbidities. Similarly, the presence of atopic disease was noted in more than 50% of the EoE cases.

The most frequently noted symptom by non-EoE-EGID patients was abdominal pain (*n* = 22), with diarrhea (*n* = 14), nausea/vomiting (*n* = 10), abdominal fullness and discomfort (*n* = 5), dysphagia (*n* = 3), heartburn (*n* = 1 case), and anorexia (*n* = 1) also reported as the chief complaint. As for the patients with EoE, the most frequently reported symptom was dysphagia, followed by heartburn and nausea/vomiting.

In laboratory testing, peripheral blood eosinophilia was the only specific abnormality frequently found in cases of non-EoE-EGID, with 19 (68%) of the 28 showing peripheral eosinophilia greater than 0.5 × 10^9^/L. In contrast, only 10% of the patients with EoE were found to have peripheral eosinophilia. Also, eight (29%) patients with non-EoE-EGID had elevated CRP, whereas this was not noted in any of the EoE patients. Six (21%) non-EoE-EGID patients demonstrated a decreased peripheral blood albumin level, all of whom had small intestine involvement including the duodenum. Additionally, decreased hemoglobin concentration was observed only in the non-EoE-EGID cases (Figure 4).

Lesions in single or multiple gastrointestinal segments between the stomach and colon were noted in all patients with non-EoE-EGID, with the duodenum most frequently involved in 15 (54%) patients, and the gastric antrum was the second most frequently involved segment (Figure 5). Nine non-EoE-EGID cases (32%) showed esophageal involvement in addition to gastrointestinal lesions. Different from esophageal lesions (longitudinal furrows in five cases, rings in two cases, white exudate in two cases), lesions found in the gastrointestinal tract by endoscopy were neither specific nor characteristic of non-EoE-EGID cases (Table 1). Non-specific edema (*n* = 11), erythema (*n* = 9), ulcers/erosion (*n* = 6), white plaque (*n* = 4), granularity/nodules (*n* = 2), and mucosal friability (*n* = 1) were shown in the endoscopic results of the 28 non-EoE-EGID cases. All 20 of the EoE cases had longitudinal furrows, mainly in the lower and middle part of the esophagus, while 12 cases also had white plaque and 10 cases had rings. Longitudinal furrows and rings are characteristic of EoE and good indications for endoscopic identification of the disease, different from endoscopic diagnosis for non-EoE-EGID cases.

Peak eosinophil counts identified in an HPF (×400) as part of histopathological examinations are shown in Figure 6. Eosinophils infiltrating each involved mucosal segment are shown as dots. However, cut-off levels for a diagnosis of non-EoE-EGID have yet to be unanimously determined by gastrointestinal pathology specialists. Therefore, we used cut-off levels for a diagnosis of EGID that were based on findings of our previous investigation of normal eosinophil infiltration levels in a large number of healthy Japanese adults, as follows: >15 in the esophagus, >20 in the stomach and duodenum, >30 in the small intestine, >50 in the right-side colon, and >30 in the left-side colon [30]. This study is the only available study investigating the normal cut-off level of gastrointestinal mucosal eosinophil infiltration in Japanese normal adults [30].

For treatment of non-EoE-EGID, prednisolone is most frequently used, followed by montelukast. Eighteen of the cases in the present case series were treated with prednisolone, all of whom showed a response, although the majority of cases also demonstrated recurrence when the prednisolone dose was reduced. Montelukast was given in 14 cases, with remission state occurring in five of the cases simply from that administration. In addition, treatments with a proton pump inhibitor (*n* = 6), antihistamine H1 blocker (*n* = 5), topical glucocorticoid (*n* = 2), azathioprine (*n* = 2), and omalizumab (*n* = 1) were given. Also, an elimination diet for eliminating allergens was used in one case and surgery for ileal stenosis in one case.

PPIs including vonoprazan were administered to all the cases with EoE. Only five of the 20 EoE patients (25%) required additional administration of topical glucocorticoid, fluticasone, or budesonide. Invasive treatment such as balloon dilatation was not necessary in any of these cases.

### 3.2. Discussion on the Differences between EoE and Non-EoE-EGID

The 28 non-EoE-EGID and 20 EoE cases in this series were compared, with differences and similarities shown in Table 3. First, heterogeneity was more often seen in non-EoE-EGID. EoE was mainly found in relatively young males, while non-EoE-EGID patients had a wide age range with a nearly 1:1 male/female ratio. Both diseases were frequently accompanied by atopic disease. Severity was higher and abnormal laboratory test results were more frequently found in the non-EoE-EGID group (Figure 2). Additionally, intractable abdominal pain and severe diarrhea, as well as hypo-albuminemia were only noted in the patients with non-EoE-EGID. High-grade peripheral eosinophilia was also found in some non-EoE-EGID cases but none of the EoE cases. A diagnosis of EoE is generally not difficult because of the presence of endoscopically identifiable specific abnormalities, while an endoscopic diagnosis of non-EoE-EGID can be complex and random multiple biopsy sampling is necessary for accuracy. Mucosal edema is a frequently reported finding of pediatric EoE patients [65]. In our case series, edema is not reported by endoscopists in adult cases with EoE. Endoscopists may report only specific findings including longitudinal furrows and they cannot report mucosal edema, due to its non-specificity for the diagnosis of EoE.

Treatment strategies for EoE have been approved by investigators and clinicians, with proton pump inhibitors effective in over 60% of affected cases. The effectiveness of topical glucocorticoid treatment has also been established for EoE, while its role for non-EoE-EGID has yet to be fully confirmed. Effective control of disease activity in EoE patients is much easier than in those with non-EoE-EGID. Even when using a combination of treatments, it is not necessarily easy to control disease activity in non-EoE-EGID patients. Anti-IgE antibody administration has been found to be effective for some non-EoE-EGID cases, including one patient in the present case series, while it has been shown to be not beneficial for treatment of EoE. Thus, while EoE and non-EoE-EGID have several differences, they also demonstrate similar clinical characteristics.

## 4. Conclusions

Non-EoE-EGID is likely a heterogenous gastrointestinal allergic disease, although it has not been fully clarified. Notably, Th2-type immune reaction and ILC2 innate immunity are key mechanisms involved in its development. With the increasing trend of various allergic diseases, greater numbers of reports regarding non-EoE-EGID are being presented, which suggests an increasing prevalence of the disease. For a diagnosis of non-EoE-EGID, the presence of gastrointestinal symptoms and abnormally dense infiltration of eosinophils in gastrointestinal tissue are key factors, although the upper limits of physiological eosinophil infiltration in each segment of the gut have not been fully determined. Since endoscopy is not adequately sensitive to detect eosinophil infiltration and lesions identified with that are not specific, it is necessary to obtain multiple biopsy samples from various segments of the gut. A diagnosis of non-EoE-EGID can only be possible after exclusion of a long list of other diseases that should be ruled out first. Moreover, an effective treatment strategy has yet to be fully constructed and new treatment options are under development. Additional research will be necessary to provide more effective and useful treatment for non-EoE-EGID patients.

## Figures and Tables

**Figure 1 biomolecules-13-01417-f001:**
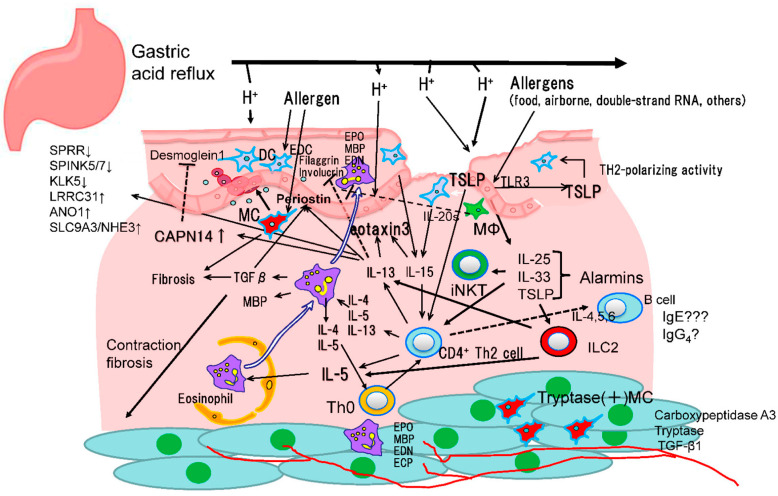
Suggested pathogenesis of EoE.

**Figure 2 biomolecules-13-01417-f002:**
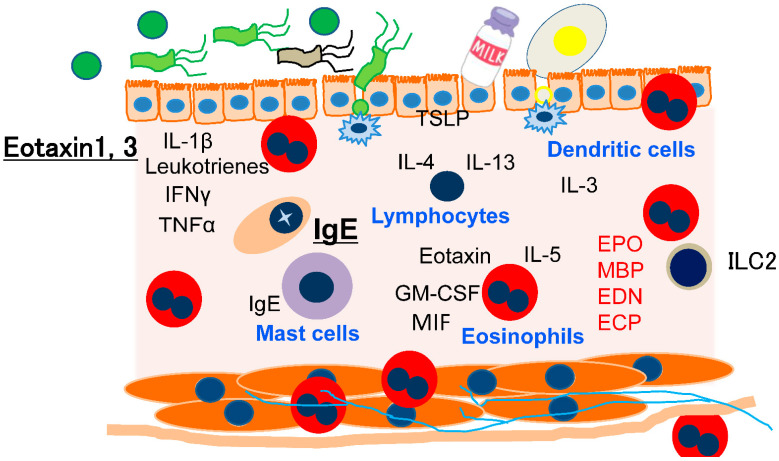
Suggested pathogenesis of non-EoE-EGID. As compared with EoE, knowledge regarding the pathogenesis of non-EoE-EGID cases is limited.

**Figure 3 biomolecules-13-01417-f003:**
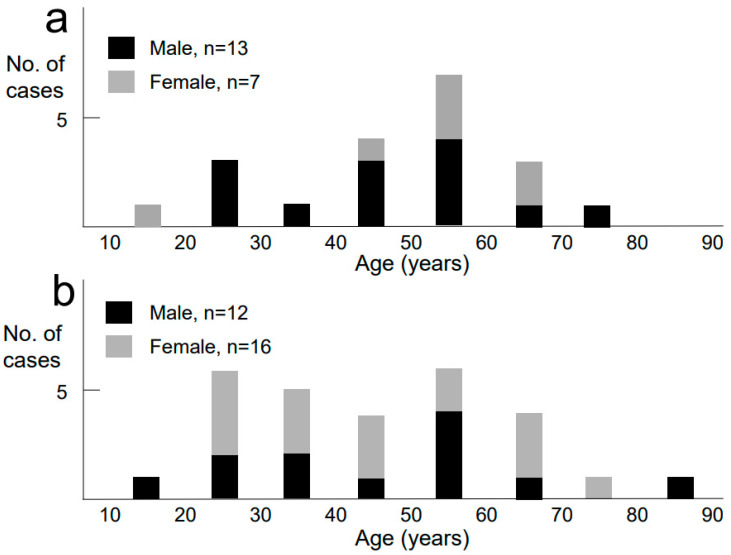
Case series of 28 non-EoE-EGID and 20 EoE patients. Age and gender distribution for: (**a**) EoE and (**b**) non-EoE-EGID. Patients with EoE were mainly in their 40s and 50s, with a male preponderance. Patients with non-EoE-EGID had a wider age range with a female preponderance.

**Figure 4 biomolecules-13-01417-f004:**
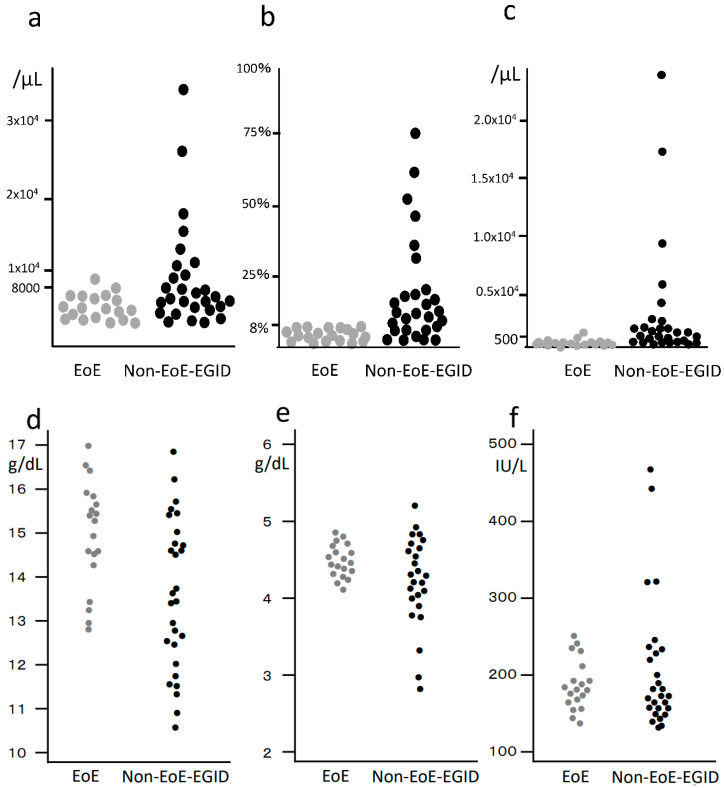
Case series of 28 non-EoE-EGID and 20 EoE patients: (**a**) Peripheral blood leucocyte count; (**b**) eosinophil fraction in leucocytes; (**c**) absolute eosinophil count. Elevated numbers of leucocytes and eosinophils were more frequently found in the non-EoE-EGID cases, with 68% of the non-EoE-EGID cases showing an elevated peripheral eosinophil count as compared to only 10% of the EoE cases; (**d**) peripheral blood hemoglobin concentration; (**e**) serum albumin; (**f**) LDH in EoE and non-EoE-EGID patients. Some of the non-EoE-EGID cases showed decreased hemoglobin and albumin concentrations, and increased LDH level, while, with respect to these measurements, there were no abnormal values among the EoE cases.

**Figure 5 biomolecules-13-01417-f005:**
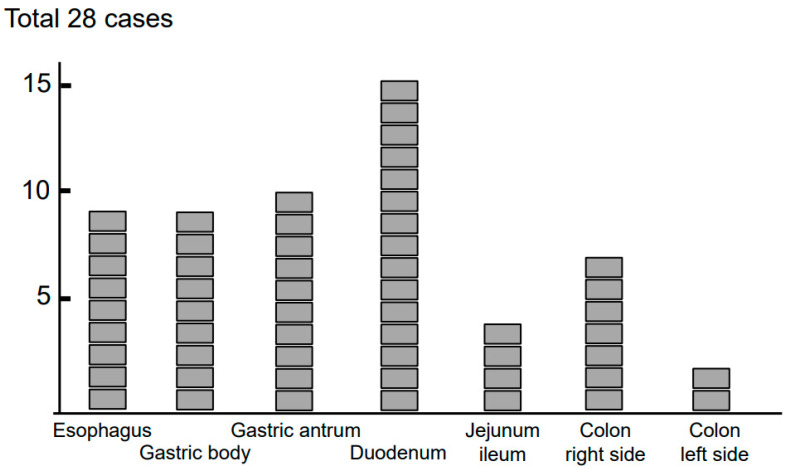
Case series of 28 non-EoE-EGID patients. Gastrointestinal tract involvement with eosinophilic inflammation was noted in patients with non-EoE-EGID. When more than two lesions were found in a single patient, each of the involved gastrointestinal segments was counted. The duodenum was the most frequently involved segment in non-EoE-EGID cases.

**Figure 6 biomolecules-13-01417-f006:**
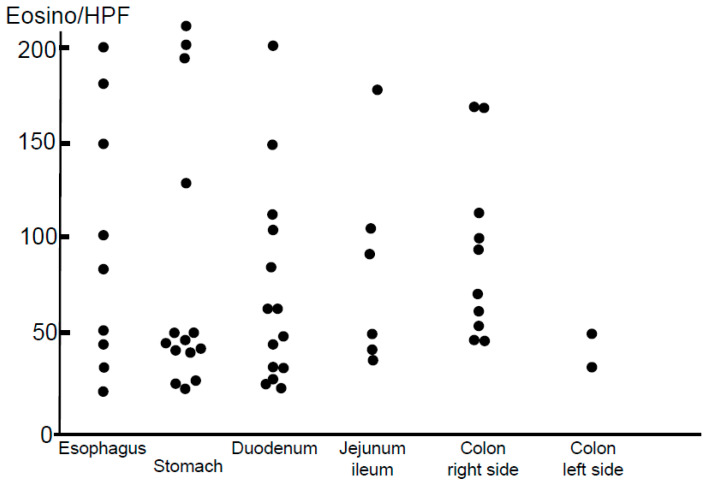
Peak numbers of eosinophils infiltrating different segments of the gastrointestinal tract in patients with non-EoE-EGID are shown.

**Table 1 biomolecules-13-01417-t001:** Diseases to differentiate from non-EoE-EGID.

Irritable bowel syndrome
Functional dyspepsia
Crohn’s disease
Ulcerative colitis
Celiac disease
Eosinophilic granulomatosis with polyangiitis (EGPA)
Henoch-Schoenlein purpura
Infectious enterocolitis
Parasitic infection
Eosinophilic leukemia
Hyper-eosinophilic syndrome
Radiation enterocolitis
Ischemic colitis
Ischemic enteritis
Malignant lymphoma
Non steroidal anti-inflammatory drug (NSAID)-related enteropathy
Pollen food allergy syndrome
Others

**Table 2 biomolecules-13-01417-t002:** Characteristics of EoE and non-EoE-EGID patients.

	EoE	Non-EoE-EGID
Number	20	28
Male/Female	13/7	12/16
Mean age(years)	47.4	44.0
Allergic disease		
Asthma	2	6
Allergic rhinitis	9	5
Food allergy	5	4
Eosinophilic chronic rhinosinusitis	0	3
Atopic dermatitis	4	2
Drug allergy	3	2
Eosinophilic otitis media	0	1
Symptoms		
Dysphagia	10	3
Heartburn	6	1
Chest pain	2	0
Abdominal pain	1	22
Diarrhea	0	14
Nausea/vomiting	3	10
Abdominal fullness/discomfort	1	5
Anorexia	0	1
Endoscopic findings		
Longitudinal furrows	20	5
White plaque	12	4
Rings	10	2
Edema	0	11
Erythema	0	9
Ulcers/erosion	0	6
Granularity/nodules	0	2
Friability	0	1
Therapy		
Prednisolone	0	18
Montelukast	2	14
PPI/vonoprazan	20	6
Histamine H1 blocker	0	5
Topical glucocorticoid	5	2
Azathioprine	0	2
Omalizumab	0	1
Surgery	0	1

**Table 3 biomolecules-13-01417-t003:** EoE and non-EoE-EGID similarities and differences.

	EoE	Non-EoE-EGID
Pathogenesis	Th2, ILC2	Th2, ILC2, IgE
Heterogeneity of patients	homogenous	heterogenous
Age at onset	40–50 years	Any age
Gender	F/M = 1/2	F/M = 1.3/1
History of atopic diseases	>50%	>50%
Symptom	Dysphagia, heartburn, vomiting	Abdominal pain, diarrhea, nausea/vomiting
Severity	Mil	Occasionally severe
Peripheral eosinophilia	Rare	Frequent
Hypo-albuminemia	Rare	Occasional
Endoscopic imaging	Characteristic (longitudinal furrows, rings, white plaque, etc.)	Not characteristic (edema, erythema, erosion, ulcer, granularity, etc.)
Criteria for pathological diagnosis	International consensus	No international consensus
Treatment	standardized	not standardized

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
