# Peer review of "(untitled)"

_biomolecules, 2023, doi:10.3390/biom13091417_

Round 1
Reviewer 1 Report
Attached

None
Author Response
Thank you very much for reviewing our manuscript and sending constructive comments. We have revised our manuscript according to your comments. We believe your comments improved our manuscript significantly. Thank you again for your help. The individual responses to your comments are outlined below.
a. Thank you for your comment. The full text of abbreviations are added in the text and title.
b. In the section of laboratory testing, the sentence suggeesting the utility of fecal calprotectin was added on lines 130-131. A new reference 24 was also added.
c. References 30-32 were added to the sentence "cut-off levels for diagnosis of non-EoE-EGID differ among investigators and indicarte the need for diagnostic markers other than simple density of eosinophil infiltration" according to the reviewer's suggestion. The cut-off levels used in the case series are more precisely described on lines 390-392.
d. According to the reviewer's suggestion, we have added the sentences on lines 213-231. In addition, we have added new references 45,46,47 in the revised manuscript.
e. Thank you for the suggestion, we have added new sentences on line 288-293.
Reviewer 2 Report
This paper is a review of general concepts on EGIDS and a description of 28 EGIDS cases in a tertiary centre in Asia. The paper is largely well written, flows well, and the message is effectively delivered.
I only have a few major and minor comments that need to be addressed.
MAJOR
1. In the “case series” section, the authors should report how did they select EoE and EGID patients for inclusion. Were these consecutive patients?
2. In the methods the authors should describe all the procedures that were performed and if these were performed in all patients. (e.eg, blood test for all patients, upper and lower endoscopy for all patients). The authors should also describe if non-EoE EGIDS had been ruled out in patients with EoE.
3. Had all non-EoE EGIDS been biopsied in all the gastrointestinal segments? This should be reported and described.
4. The authors should avoid describing subjective opinions in the results section, for example, but not only, “Even when using a combination of treatments, it is not necessarily easy to control disease activity in non-EoE-EGID patients.” I would stringly recommend adding a separate discussion paragraph on this.
MINOR
1. Page 2, line 48. I do not feel that only 28 cases of EGIDS can “clarify the clinical characteristics of Asian patients” with EGIDS. Please remove this statement.
2. Page 2. The description of the pathogenesis of EoE should be implemented with information regarding the possibility of pollen sensitization as a trigger of EoE, as this has just been demonstrated (https://pubmed.ncbi.nlm.nih.gov/37307575/).
3. Laboratory testing paragraph. The reference on laboratory testing should be updated as novel diagnostic/follow-up biomarkers are becoming available (https://www.mdpi.com/2075-4418/13/17/2806)
Minor polishing required
Author Response
Thank you very much for reviewing our manuscript and sending constructive comments. We have revised our manuscript according to your comments. We believe our review article has been improved significantly by revising it according to your comments. Thank you again for your help. Our point-by-point responses are outlined as follow.
MAJOR
- Thank you for your clarification. We did not select enrolled patients in our case series. All the consecutive patients during the inclusion period are included in the study. To clearly show this, we have added one paragraph concerning the selection of enrolled cases. Page 8, Line 292-302.
- According to the reviewer's suggestion, we have described all the diagnostic procedure employed in this case series. We have also added the sentence describing that, in patients with EoE, the presence of non-EoE EGIDS has been ruled out. Page 8, Line 292-302.
- Thank you for the clarification. Since this is a retrospective study, not all the gastrointestinal segments of all the enrolled cases were investigated histologically. Upper gastrointestinal endoscopy was done on all the enrolled cases and histological study was done on esophageal, gastric, and duodenal mucosa in all the enrolled cases. Colonoscopy was performed only when the patients have symptoms or signs suggesting lower gastrointestinal tract involvement. This information has been added in the revised manuscript. Page 8, Line 292-302.
- Thank you for the suggestion. We have deleted our opinion from the results section and put them in the section "Discussion on the difference of EoE and non-EoE-EGID. Page 13. Line 383-409.
MINOR
- According to the reviewer's suggestion, we have deleted the words "clarify the clinical characteristics of Asian patients" from revised manuscript.
- According to the reviewer's suggestion, we have added a new reference 12. We have also added one sentence concerning pollen sensitization on Page 2, Line 70-71.
- According to the reviewer's suggestion, we have added a new reference 23. We have also added one sentence concerning non-invasive diagnostic procedures on Page 4, Line125-126.
Reviewer 3 Report
Line 126: needs a reference
line 158: "As a result, cut-off levels for diagnosis of non-EoE-EGID differ among investigators" Not only differences in microscopes but the OBSERVERS.
Please see the reference:
in Table 2 is shown that edema is more frequent in non-Eos contrarywise of what was in EoS. I wish to have more discussion on that because in children, edema is a common histopathological finding in patients with EoE . The paper "Endoscopic and histological characteristics in patients with eosinophilic esophagitis responsive and non-responsive to proton pump inhibitors. J Pediatr (Rio J). 2020;96(5):638-643" . may help the authors to improve discussion.
Figure 4: please correct values in veritcal axis.
Figure 6 : legend " Case series of 28 non-EoE-EGID and 20 EoE patients. Peak numbers of eosinophils infil- 356 trating different segments of the gastrointestinal tract in patients with non-EoE-EGID are shown..
legend: Peak numbers of eosinophils infil- 356 trating different segments of the gastrointestinal tract in patients with non-EoE-EGID are shown
Author Response
Thank you very much for reviewing our manuscript and sending constructive comments and suggestions. We have revised our manuscript according to the reviewer's suggestion. We believe our manuscript has been improved significantly with your help. The individual point-by-point response to your comments are as follow.
- According to the reviewer's suggestions, new references 23 and 24 are added on page 4, line 129.
- Thank you for the comment, we have added new sentence concerning the interobserver variance of eosinophil counting and a new reference 31 on page 4,5, line 158-159.
- Thank you for the constructive suggestion, we have added a new reference 62 and sentences concerning the endoscopically identified mucosal edema on our discussion section on page 13, line 395-399.
- Thank you for your help, the description on vertical lines in Figure 4 has been corrected.
- Figure legend of Figure 6 has been changed according to the reviewer's suggestion.
Reviewer 4 Report
This is an excellent review article focusing on non-EoE-EGID. I have some minor comments.
Line 39. The word “reprted” should be corrected.
Line 63. The word “that” may need to be changed to “than.”
Line 65. The word “air-born” needs to be changed to “air-borne.”
Figure 1. Why are there white and black arrows?
Figure 2. Mast cells and ILC2 should be differently demonstrated.
Lines 90, 192, and 273. Is the section number “1” necessary?
Lines 260 to 272. Is anti-IL-5 or anti-IL-5Ra antibody effective?
Line 313. The word “great” will need to be changed to “greater.”
Figure 4, a and c. Non-English characters are used as Y-axis.
Lines 343 and 356. The phrase “and 210 EoE” is not necessary since the figures do not include patients with EoE.
Line 408. The word “effective” could be changed to “effective and useful”, since patients will hope drugs with equal effects and less side effects compared to prednisolone.
Nothing especially.
Author Response
Thank you very much for reviewing our manuscript and sending constructive comments. We have revied our manuscript according to the reviewer's suggestions. We believe our review article has been improved significantly due to your help. The individual point-by-point responses are outlined as follow.
- Thank you for the correction. The word "reprted" has been corrected to "reported" on line 39.
- The word "that" has been corrected to "than" on line 62.
- The word "air-born" has been corrected to "air-borne" on line 65.
- In Figure 1, white and black arrows show the same meaning. Therefore, we have changed all the white arrows in Figure 1 to black ones. Thank you for your suggestion.
- In Figure 2, ILC2 was differently illustrated from mast cells in the revised manuscript. Thank you for your suggestion.
- The section numbers "1" have been deleted from the revised manuscript on lines 90, 192, and 273.
- There is no confirmed evidence to the best of our knowledge that shows the effectiveness of anti-IL-5 or anti-IL-5Ra antibody for the treatment of non-EoE-EGID. Therefore, we did not include anti-IL-5 and anti-IL-5Ra antibodies on page 7, lines 260-272.
- The word "great" on line 313 has been changed to "greater" in the revised manuscript.
- Non-English characters on Figure 4 have been changed to the correct ones. Thank you for your help.
- According to the reviewer's suggestion, the phrase "and 20 EoE" has been deleted from legend of Figure 5 and 6. Thank you for the correction.
- The word "effective" on line 408 has been changed to "effective and useful" according to the reviewer's suggestion.
Round 2
Reviewer 1 Report
None
None